# Fractures in Children Due to Firearm Activity

**DOI:** 10.3390/children10040651

**Published:** 2023-03-30

**Authors:** Randall T. Loder, Taylor Luster

**Affiliations:** 1Department of Orthopaedic Surgery, Riley Children’s Hospital, Indiana University School of Medicine, Indianapolis, IN 46202, USA; 2Division of Student Affairs, Indiana University School of Medicine, Indianapolis, IN 46202, USA

**Keywords:** firearm, fracture, children, demographics, spine, extremity, hospital admission

## Abstract

The purpose of this study was to investigate fracture patterns due to pediatric firearm injuries. The data used was from the US Firearm Injury Surveillance Study 1993–2019. Over these 27 years, there were 19,033 children with fractures due to firearm activity with an average age of 12.2 years; 85.2% were boys and the firearm was a powder type in 64.7%. The finger was the most common fracture location, while the tibia/fibula was the most common location for those admitted to the hospital. Children ≤ 5 years of age sustained more skull/face fractures; most spine fractures occurred in the 11–15-year age group. The injury was self-inflicted in 65.2% of the non-powder and 30.6% of the powder group. The injury intent was an assault in 50.0% of the powder and 3.7% of the non-powder firearm group. Powder firearms accounted for the majority of the fractures in the ≤5 and 11–15 year-olds, non-powder firearms accounted for the majority of the fractures in the 6–10 year-olds. Injuries occurring at home decreased with increasing age; there was an increase in hospital admissions over time. In conclusion, our findings support the need for safe storage of firearms in the home away from children. This data will be helpful to assess any changes in prevalence or demographics with future firearm legislation or other prevention programs. The increasing acuity of firearm-associated injuries seen in this study is detrimental to the child, impacts familial wellbeing, and results in significant financial costs to society.

## 1. Introduction

Injuries due to firearms are a significant health burden [1,2,3]. Deaths attributed to firearms in the US population are equivalent to those from motor vehicle crashes and falls [2]. Additionally, firearm injuries result in significant societal costs, including both financial damage and loss of human life/work [3,4,5]. These injuries do not occur solely in adults, but also in children [6,7,8,9]. Pediatric firearm injuries result in significant costs to society [6,10,11,12]. Deleterious firearm injuries in children also result in emotional trauma for families due to the loss or injury of a child, financial burden [6,7,8,9], and rehabilitation costs, as 8.4% of children with firearm injuries are discharged to a rehabilitation facility [13].

Pediatric orthopaedists are often called upon to care for a child with a fracture arising from a firearm injury [14]. There are several studies regarding firearm injury fracture patterns and associated demographics in children [14,15,16,17,18,19,20,21,22,23,24]. These studies, although informative, are limited in scope by various parameters. For example, some studies only include a certain geographic area [16,19,20,23,24,25], only those admitted to the hospital [14,19,20,24], short time periods [16,17,23,25], or difficult fractures [15,26]. Most are either limited to powder firearms [14,15,16,17,19,20,23,24,25] or non-powder firearms [18] and very few studies mention spine fractures [14,19]. The difference between powder and non-powder firearms is the source of the energy used to project the bullet out of the gun. Powder firearms use the gases from the explosion of the gunpowder, while non-powder firearms use compressed air or other gases. One study covers an entire nation, albeit a small one (Jamaica) [17]. 

It was the purpose of this study to concentrate on the demographics and fracture patterns of injuries due to firearms in children over a quarter of a century using a national emergency department (ED) visit the database. This will include both those treated and released as well as admitted to the hospital, all areas of the country, and both powder and non-powder firearms. The strength of this study is that it will provide a large overview of pediatric fractures from injuries due to firearm activity, be useful as baseline data for future studies regarding these injuries, and perhaps serve as a guide for injury prevention programs.

## 2. Materials and Methods

### 2.1. Data Source

The data for this study was obtained from the Inter-University Consortium for Political and Social Research Firearm Injury Surveillance Study 1993–2020 (ICPSR 38574) (https://www.icpsr.umich.edu/web/NACJD/studies/38574, accessed on 18 December 2022) collected by the National Electronic Injury Surveillance System (NEISS). The NEISS, a branch of the US Consumer Product Safety Commission, collects data from a probability sample of hospitals in the United States and its territories that have at least six beds and an ED. The sample contains five strata, four based on size (the total number of emergency room visits reported by the hospital and are small, medium, large, and very large) and one stratum consisting of children’s hospitals. There are ~100 hospitals in the NEISS, and this number varies slightly from year to year. Patient information is collected daily from each NEISS hospital for every patient treated in the ED due to an injury associated. The ICPSR data set consists of any patient seeking care in the ED for firearm-related injury, regardless of activity involved during the injury (e.g., hunting, committing a crime, suicide, assault), and whether the patient had been shot by the firearm or injured in some other way (e.g., a skull/face fracture from being pistol-whipped, a clavicle fracture from a rifle recoil, etc.). Further details regarding the acquisition of the ICPSR/NEISS data and guidelines for use of such data can be accessed from their respective websites (ICPSR—www.icpsr.umich.edu, NEISS—www.cpsc.gov/library/neiss.html, accessed on 18 December 2022). This study of publicly available de-identified data was considered exempt by our local Institutional Review Board.

The data from 1993 through 2020 was downloaded from the ICPSR website; however, due to changes in injury patterns associated with the COVID pandemic, those data from the year 2020 were excluded [27,28,29,30,31,32], including firearm injuries [33,34,35]. The data includes age, sex, race, type of firearm, the perpetrator of the injury (e.g., self, stranger, etc.), intent of injury (unintentional, assault, suicide, law enforcement), anatomic location of the injury, incident locale (home, street/highway, etc.), disposition from the ED, involvement of drugs/crime/fight/argument in the incident, and was the patient shot or not with the firearm. Race was classified as White, Black, Amerindian (Hispanic and Native American), and Asian [36]. We limited our study to those 15 years or younger. The 16-year-old age limit was used as most patients ≥ 16 years of age demonstrate adult fracture patterns and 16 is when the transition into adult activities, such as driving, begins. 

Fractures were identified using two different methods. The first method was to identify fractures using the NEISS diagnosis code of 57, the code for a fracture. As the most serious injury is used to determine the diagnosis by the NEISS, many patients sustained fractures along with more serious injuries, such as a pneumothorax. In order to identify additional fractures, the database column CMTX was utilized; this column is a description of the event where patient identification has been purged by the CDC and CPSC. The database was searched for other fractures using the FIND command in Microsoft Excel™ (Microsoft^®^ Office 365 Apps for enterprise) using the terms fx and frac. To further search for spinal injuries the database was searched using the terms spinal, verte, paral (for paralyzed), spine, cervical, thoracic, parapl and quadri (for paraplegia and quadriplegia), and each vertebral level (e.g., C1, C-1) from C1 through S4. The anatomic location of the fracture was then determined and ranked from the most serious to least serious when there was more than one fracture. Fractures of the spine were classified as the most severe, followed by the skull, pelvis, sternum/ribs, and, finally, the long bones of the appendicular skeleton beginning from proximal to distal respectively. The fracture locations were condensed into 5 major groups: spine, upper extremity, lower extremity (including the pelvis), skull/face, and rib(s). 

Patients were separated into 3 groups by age (≤5, 6 through 10, and 11 through 15 years). The 27 years covered in this study were separated into 3 equal groups of 9 years (1993 through 2001, 2002 through 2010, and 2010 through 2019). Finally, we wished to study differences in fracture patterns by assessing if the shooting was a drive-by shooting or not; drive-by shootings were found by searching the database using the FIND command for the terms driveb, drive-by, drive-b, driveth, drive th, and drive-th.

### 2.2. Statistical Analysis

Statistical analyses were performed with SUDAAN 11.0.01™ software (RTI International, Research Triangle Park, North Carolina, 2013) which accounts for the weighted, stratified nature of the data, giving an estimated number of ED visits, along with 95% confidence intervals (CI) of the estimate N denoted by brackets [ ]. When the actual number of patients (n) is <20, the estimated number (N) becomes unstable and should be interpreted with caution; thus, we report both the n and N. Analyses between groups of continuous data were performed with the *t*-test (2 groups) or ANOVA (3 or more groups). Differences between groups of categorical data were analyzed by the χ^2^ test. For all analyses a *p* < 0.05 was considered statistically significant.

## 3. Results

Over the 27-year period of 1993 through 2019, there were 111,796 actual ED visits for injuries due to firearms, resulting in an estimated 3,359,809 [2,956,755, 3,744,864] ED visits after appropriate statistical analysis using the weighted data. Of these 3.36 million ED visits, an estimated 434,458 [356,526, 526,747] (13.0%) were in those <16 years of age. Of these 434,458 ED patients an estimated 19,033 [15,814, 22,852] (4.4%) sustained fractures. Therefore, these 19,033 ED visits comprise this study. From here on, only the estimated number (N) will be given in the manuscript text and used in the figures; both the actual (n) and estimated number (N) of ED visits are given in the Tables. 

The average was 12.2 years; 85.2% were boys and the firearm was a powder type in 64.7%. All the data for the many different variables are given in Table 1.

There were 19,370 fractures in these 19,033 patients. The exact number of fractures was known in 19,011 patients and was 1 in 18,640 (98.05%), 2 in 365 (1.92%), and 3 in 6 (0.03%). The detailed anatomic distributions for all patients as well as two separate groups of those released from the ED and those admitted to the hospital are shown in Table 2. 

The finger was the most common fracture location for patients both overall and released from the ED. The tibia/fibula was the most common fracture location for those admitted to the hospital. For those released from the ED, the upper extremity was the most common location (61%) (Figure 1a) and for those admitted to the hospital, it was the lower extremity (45%) (Figure 1b).

There were very few Asian children, children with isolated rib fractures and those having more than one fracture. Thus, we excluded Asian children, those with an isolated rib fracture, and those with more than one fracture. All the subsequent analyses were performed with these exclusions.

### 3.1. Analyses by Fracture Group

Notable differences (Table 3) include children ≤ 5 years of age sustained more skull/face fractures (Figure 2a). Most spine fractures, while rare, occurred in the 11–15-year age group. Fractures of the upper extremity accounted for 58.6% of all the fractures in the 6–10 age group. Nearly all spine fractures (98%) were associated with powder firearms (Figure 2b). Patients with lower extremity fractures were more commonly admitted to the hospital compared to those with upper extremity fractures; the few deaths occurred exclusively in those with fractures to the skull/face (Figure 2c). While most of the patients sustained a gunshot wound (i.e., were shot), nearly all of those with spine fractures were shot, while 39% of those with skull/face fractures were not shot (Figure 2d). Other statistically significant differences existed by race, perpetrator, incident locale, and injury intent.

### 3.2. Analyses by Powder vs. Non-Powder Firearms

In addition to the differences described above by fracture location and firearm type, there were notable differences by sex, race, disposition from the ED, perpetrator of the injury, injury intent, incident locale, and age groups (Table 4). 

Boys comprised 81.3% of the powder and 92.1% of the non-powder firearm groups (*p* = 0.0024). White children accounted for 45.7% of the powder and 71.6% of the non-powder firearm group (*p* = 0.0005) (Figure 3a). Of those patients with fractures due to powder firearms, 42.5% were admitted to the hospital, while only 6.0% of those due to non-powder firearms were admitted (*p* < 10^−4^). The injury was self-inflicted in 65.2% of the non-powder and 30.6% of the powder group (Figure 3b) (*p* < 10^−4^). The injury intent was an assault in 50.0% of the powder and 3.7% of the non-powder firearm group (Figure 3c) (*p* < 10^−4^). While the injuries occurred at schools or places of recreation in only 6.8% of all the patients (Table 1), those fractures which occurred at schools or places of recreation were due to powder firearms in 92.9% (Figure 3d) (*p* < 10^−4^). Although there was minimal difference in the average age between the two groups (12.5 years—powder, 11.7 years—non-powder, *p* = 0.58), there was a significant difference between the three age groups. Powder firearms accounted for the majority of the fractures in the ≤5 and 11–15 yea-olds, non-powder firearms accounted for the majority of the fractures in the 6–10-year-old group (Figure 3e) (*p* = 0.006). No differences were observed in the patient being shot or not shot by firearm type.

### 3.3. Analyses by Being Shot or Not Shot

In addition to the differences by major fracture, groups noted above, there were notable differences by incident locale and disposition from the ED (Table 5). 

Children who sustained injuries at schools and recreational facilities were less likely to be shot compared to other places (Figure 4a). Examples would be a clavicle fracture sustained from a rifle recoil while doing target practice, or a nasal fracture to a participant in marching band/color guard activities. All deaths and nearly all of those admitted to the hospital had been shot (Figure 4b), while 26% of those released from the ED were not shot (*p* < 10^−4^) and experienced injury from the firearm in a different way. There were no differences between those shot or not shot by firearm type or perpetrator of the injury; however, there were differences by race and injury intent. White children comprised 71.7% of those not shot and 51.2% of those shot (*p* = 0.036); the injury was unintentional in 76.1% of the shot group and 56.2% of the shot group (*p* = 0.0005).

### 3.4. Analyses by Disposition from the ED

In addition to the differences by fracture location, firearm type, and being shot or injured in another way, those admitted to the hospital from the ED (Table 6) were less commonly White (Figure 5a), less frequently injured themselves (Figure 5b), and more commonly injured due to an assault (Figure 5c). The rate of hospital admissions increased over time (*p* = 0.004) (Figure 5d). 

### 3.5. Analyses by Age Groups and Drive-by Shootings

In addition to the difference by firearm type previously noted, the percentage of injuries occurring at home decreased with increasing age (63.3% < 5 years, 54.5% 6 to 10 years, and 32.4% 11 to 15 years of age) (Figure 6). No other significant differences existed between the different age groups. Regarding those injured in drive-by shootings, differences existed by race, firearm type, and perpetrator of the injury. Drive-by shooting patients were 11.2% White, 62.5% Black, and 26.3% Amerindian; non-drive-by shooting patients were 57.0% White, 31.5% Black, and 11.5% Amerindian (*p* = 0.008). The involved firearm was a powder firearm in 88.9% of the drive-by and 63.8% of the non-drive-by patients (*p* = 0.022). The perpetrator was unknown in 55.8%, a stranger in 21.3%, and not seen in 23.1% of the drive-by shootings; the perpetrator in the non-drive-by shootings was unknown in 23.1%, a stranger in 7.2%, themselves in 44.6%, a friend/acquaintance in 8.6%, another relative in 7.1%, and not seen in 9.3% (*p* = 0.008). 

### 3.6. Variations by Time

A noticeable increase in ED visits on Saturday and Sunday was observed (Figure 7a). No pattern by month (Figure 7b) or year (Figure 7c) was present.

## 4. Discussion

The findings in this study are both similar and different to other studies in the literature. As most of the studies regarding fractures in children due to firearms are due to powder firearms, we have compared our findings to the other studies (Table 7). The percentage of boys was strikingly similar for all studies; it ranged from 78 to 91% and was 81% in this series. Most of the series demonstrated more lower extremity fractures than upper extremity fractures. Of the three studies that included spine fractures, the 5.5% in this study and the 2% in that of Naranje et al. [19] are similar, in contrast to the 18.9% in the study of Blumberg et al. [14]. We have no explanation for this finding, except that only inpatients were included in the Blumberg series [14] and they included those 16 through 20 years of age. It has been shown in a previous study that those with firearm-associated spine injuries are much more common in the 15-to-34 year-old age group [42]. Carillo et al. [43] studied 19 patients with spinal cord injury secondary to gunshot wounds. The average age was 17 years with a range of 14–19 years. The fact that we excluded those over 15 years of age likely explains some of the differences between this study and that of Blumberg et al. [14]. 

The most fractured bone in this study was the finger, likely due to the inclusion of both those patients released from the ED and non-powder firearms. When looking at only those admitted to the hospital (Table 2), the most common fracture involved the tibia/fibula (17.1%) followed by the femur (14.2%). In the only other large study, that of [14], the most common fractured bone was the femur (21.2%), followed by the spine as discussed above at 18.9%, and then the tibia/fibula at 15.0%. Again, these differences are likely due to the inclusion of those children from 16 through 21 years of age in the Blumberg study [14]. Nevertheless, the numbers in this study respectively for the tibia/fibula (17.1%) and femur (14.2%) are similar to the 15.0% and 21.2% respectively for the Blumberg study [14]. In the much smaller series of 58 gunshot fractures by Naranje et al. [19] both the femur and tibia/fibula each accounted for 19% of the fractures, again very similar to the numbers in this study.

We noted that powder firearms were responsible for the majority of the fractures in the ≤5 and 11–16-year-old groups (78.7% and 67.1%) but only 44.1% for the non-powder firearm group (Figure 3e) and that the majority (63.3%) of those in the ≤5-year-old group occurred at home (Figure 6). This confirms and supports the need for firearms in the home to be safely stored and locked and away from children [44,45,46]. It has been estimated that even in 2020 that 4.6 million US children live in homes with at least one loaded and unlocked firearm [47]. The issue of gun ownership is very emotional in the US population, and in a recent study [48] gun owners with children were more likely than those without children to feel that guns make them feel more valuable to their families. Thus, acknowledging parental motivations for gun ownership is a pivotal educational component toward firearm injury prevention. However, the initial analyses did not uncover if this particular group in this study was injured unintentionally by the child or others. We, therefore, performed detailed analyses of the perpetrator and incident locale by the three age groups. In the ≤5-year-old age group, 90.8% of the fractures were self-inflicted and occurred at home; this number was 29.6% for the 6–10 and 18.4% for the 11–15 year old age groups (*p* < 10^−4^). Therefore, it can be concluded that young children are exceptionally vulnerable to accidental dislodging of an unlocked and loaded gun left at home, furthermore, emphasizing the importance of gun safety around young children.

Another interesting finding was that those injured in schools or recreational facilities had the second highest prevalence of fractures due to powder firearms (92.9%) (Figure 3d) but the least likely (36.7% compared to the overall study 80.8%) to be shot (Figure 4a). This is most likely due to the fact that, in schools, powder-type firearms are often used in color guard or other sanctioned activities. In a recent study, 43.9% of injuries due to firearms in schools occurred in the sanctioned guard or drill activities [49]. While there is understandably significant concern regarding school mass shootings in the US, only ~37% of the patients with fractures due to school-related firearm encounters were shot. Of the 1298 patients injured at schools or recreational facilities, 696 were at schools and 602 at recreational facilities. There was no difference in the number of those injured by powder and non-powder firearms between the school and recreational facilities. 

Regarding temporal factors, the patients with fractures were more likely to be injured on the weekend than on the weekday. This is understandable as school-aged children are occupied during the weekdays, reducing access to firearm activities. Tatebe et al. [50] noted that there was an increase in pediatric firearm injuries overall in Chicago. However, we noted no variation by month in this select group of children with fractures due to firearm injuries. A previous US study of temporal variation in firearm injuries [51] using an earlier version of the Research Firearm Injury Surveillance Study 1993–2008 noted a peak in September, but with many exceptions. Thus, fractures due to firearm injuries in children are likely another one of these exceptions. 

The United States has the highest rate of pediatric firearm-related injuries, specifically 10–35 times higher than other high-income countries [50]. With pediatric firearm-related fractures increasing from 1993–2019, it is important that national prevention strategies are implemented to prevent further increases in childhood morbidity and mortality relating to firearms. Tatebe et al. [50] found that 43.6% of all firearm-associated injuries occurred outside of school hours, thus providing family support, early childhood education and scheduled after-school activities could minimize the time that children are exposed to firearms. Additionally, it has been proposed [50] that access to unsecured loaded weapons needs to be minimized with increased emphasis on education regarding firearm handling.

A more interesting finding was that over time, the percentage of children admitted to the hospital for firearm-associated fractures increased (Figure 5d), in spite of the very well-known emphasis to not admit patients to the hospital in the US, where hospital admission is typically reserved for very serious injuries and/or those needing immediate surgical treatment. If hospital admission is used as an indication of injury severity, then this is a very concerning trend. If, however, it reflects perhaps more aggressive fracture fixation, then perhaps this trend could be explained. However, the vast majority of the upper extremity and many of the lower extremity fractures in children due to firearms can be treated non-operatively, with the major exception perhaps being the femur and less so the tibia/fibula. There has certainly been an increase in operative pediatric femur fracture treatment from 1993 to 2019 and, to a lesser extent, other long bone fractures [52,53,54,55]. This may explain the trend seen here. 

We compared the patterns of fractures in those associated with powder and non-powder firearms. The literature regarding non-powder firearms (i.e., BB guns, air-powered rifles) primarily focuses on overall injury patterns and does not specifically study fracture patterns. A recent study [18] used the NEISS database and excluded powder firearm injuries, while we used the Firearm Injury Surveillance Study, which is also a NEISS database, but incorporates all firearms, both powder and non-powder. In the study by Jones et al. [18] from 1990–2016, the rate of non-powder firearm injuries decreased by 47.8%, boys accounted for 87.1% of the children; BB guns accounted for 80.8% of the injuries, followed by pellet guns (15.5%), paintball guns (3.0%), and airsoft guns (0.6%). However, there was little mention of fractures with most of the focus on ocular injuries; nonetheless, fractures were most commonly associated with hospital admission. Details of fracture anatomic location were not given. In this study, 73.5% of the fractures due to non-powder firearms occurred in the upper extremity (Table 4). Of these 4934 upper extremity fractures due to non-powder firearms, 4485 (90.9%) involved the finger, 362 (7.3%) the hand, with the remaining 87 from the wrist proximal. A similar pattern was seen in the lower extremity; of the 1313 lower extremity fractures due to non-powder firearms, 820 (62.5%) involved the toes and 311 (23.7%) the foot, with the remaining 10 (0.8%) the tibia/fibula. 

There are certain limitations of the study. First is the accuracy of the NEISS data. However, previous studies [56,57], including those involving firearms, have demonstrated over 90% accuracy of NEISS data. Second, it studies only patients seen in EDs and thus those visiting urgent care centers or other outpatient clinics are not captured in this data. However, we suspect that any serious firearm injury would be seen in an ED. Third, regional-specific analyses could not be carried out due to the de-identified nature of each hospital in the NEISS sample. It would be very interesting to study differences by region [58], especially those having stricter gun control laws compared to others, but unfortunately, that is not possible due to the de-identified status of each NEISS hospital. Fourth, the number of fractures reported is likely less than the actual number for several reasons. Potential error could stem from the clerks entering the data into the comments section and inadvertently forgetting to mention a fracture when in actuality there was a fracture. In addition, a very seriously injured person coming into the ED with a major trauma likely had fractures that were overlooked and not mentioned, especially if the patient was in extremis and/or died in the ED. An additional reason is that many of the serious head injuries with brain damage from the gunshot wound would have had an open skull fracture, but it was not so coded, it was missed. The same would be for a patient with a hemo/pneumothorax, likely having a rib(s) fracture. As this is an ED-focused database, we have no information on the length of stay for those admitted to the hospital. Finally, we can not differentiate between the injuries sustained during routine recreational use (e.g., hunting, target practice) or self defense during a perceived or actual assault due to how this database is catalogued. 

A major strength of this study is that it is a national picture of pediatric fracture patterns due to firearms over a quarter of a century. It encompasses both rural and urban areas, all races, both boys and girls, and especially studies the outcome of the ED visit—treated and released, admitted, or expired while in the hospital. While these are national estimates and may not be locally applicable, they can give healthcare providers, especially ED providers, orthopaedic surgeons, and health facility administrators important information about these events. This data will also be helpful in analyzing any changes in prevalence or demographics with any future firearm legislation, for or against gun control.

Finally, what are the financial costs of this particular group of patients? The average cost of an ED visit in the US in 2020 was $1150 (https://consumerhealthratings.com/how-much-does-er-visit-cost/, accessed on 18 December 2022). The average cost for a pediatric inpatient hospital admission in US$ 2016 was $7800 (https://consumerhealthratings.com/healthcare_category/inpatient-average-cost-typical-prices-ballpark/, accessed on 18 December 2022), or $8493 for 2020 dollars using the US Consumer Price Index inflation index calculator (https://www.bls.gov/data/inflation_calculator.html, accessed on 18 December 2022). The cost for a fatality in a US ED is unknown but assuming it is equal to a hospital admission, in this study there were 13,272 children seen in US EDs for fractures due to firearm activity and released after treatment; 5671 admitted to the hospital; and 100 fatalities. This gives an estimated cost of (13,272 × $1150) + (8493 × $5631) + (100 × $5631), or $63.9 million. This is a conservative estimate, as it does not include costs for follow-up visits from an ED (which is crucial for fracture care), associated charges for imaging (cost of the radiographs, interpretation fees from the radiologist),prophylactic antibiotics, nor costs to the parents/family/society for time lost regarding employment, childcare and other issues. Finally, the estimate for pediatric hospital admission of $7800 ($8493 in 2020$) is likely very low for this particular scenario, as admissions to the hospital for pediatric orthopaedic surgical care are likely much higher than this $7800. What these actual numbers are is difficult to know. The important point is that pediatric fractures arising from firearm activity are a significant financial burden to everyone.

## 5. Conclusions

Many of the findings in this study are sobering. The increase in hospital admissions over time for firearm-associated fractures, especially in view of hospital admissions becoming more difficult to justify in the US, is concerning. If a hospital admission is a proxy of injury severity, then firearm associated fracture injuries are becoming more severe. Growing acuity of firearm-associated injuries is detrimental to the child, but also greatly impacts familial wellbeing, societal functioning, and the US-health care system financial state.

## Figures and Tables

**Figure 1 children-10-00651-f001:**
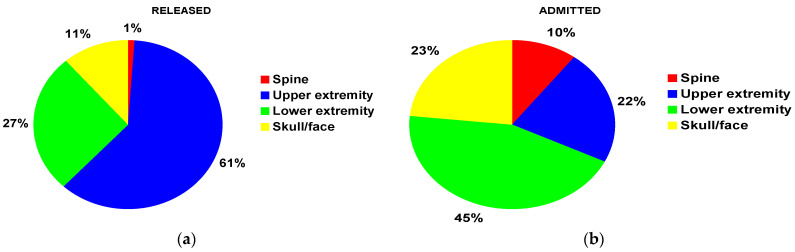
Fracture distribution by major anatomic location: (**a**) for those patients treated and released from the ED; (**b**) for those patients admitted to the hospital from the ED.

**Figure 2 children-10-00651-f002:**
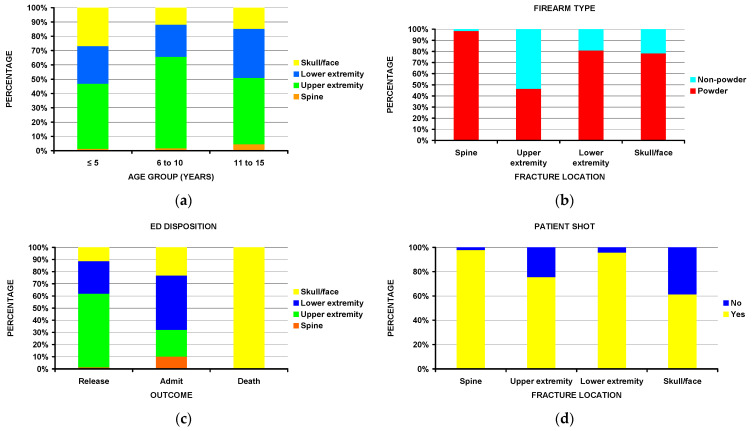
Differences between the four major fracture locations (spine, upper extremity, lower extremity, and skull/face: (**a**) by age group (*p* = 0.041); (**b**) by firearm type (*p* < 10^−4^); (**c**) by ED disposition (*p* = 0.0001); (**d**) by being shot or not (*p* < 10^−4^).

**Figure 3 children-10-00651-f003:**
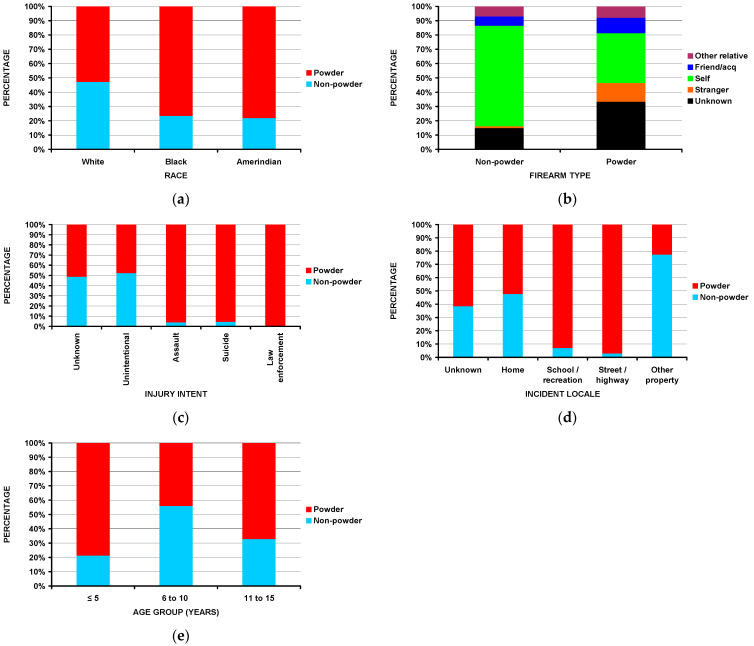
Differences between powder and non-powder firearms: (**a**) by race (*p* = 0.0005); (**b**) by perpetrator of the injury (*p* < 10^−4^); (**c**) by injury intent (*p* < 10^−4^); (**d**) by incident locale (*p* < 10^−4^); (**e**) by age group (*p* = 0.006).

**Figure 4 children-10-00651-f004:**
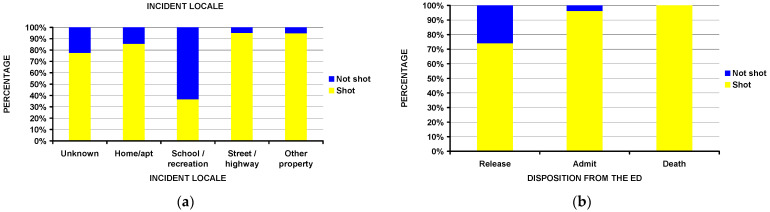
Differences between being shot or not. (**a**) By incident locale (*p* < 10^−4^); (**b**) by disposition from the ED (*p* < 10^−4^).

**Figure 5 children-10-00651-f005:**
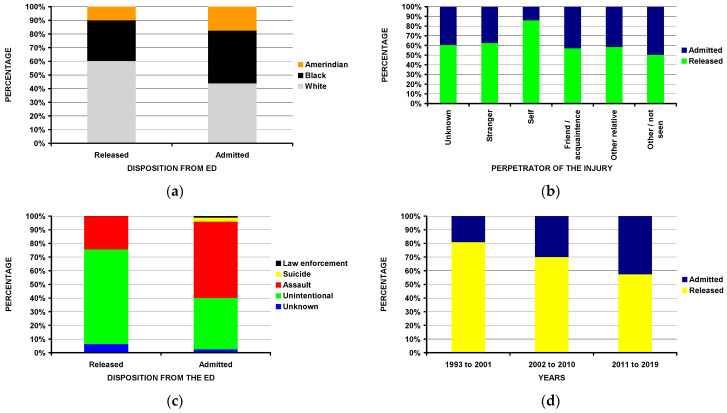
Differences by disposition from the ED: (**a**) by race (*p* = 0.011); (**b**) by perpetrator of the injury (*p* = 0.0006); (**c**) by intent of the injury (*p* < 10^−4^); (**d**) by year time span (*p* = 0.004).

**Figure 6 children-10-00651-f006:**
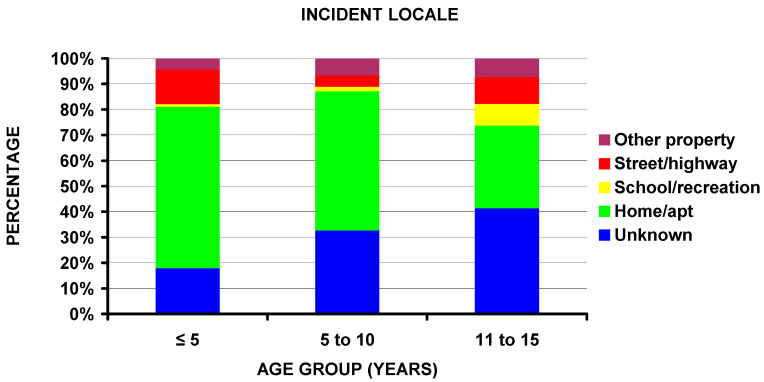
Differences in age group by incident locale (*p* = 0.037). Note the decreasing number of cases occurring at home with increasing age.

**Figure 7 children-10-00651-f007:**
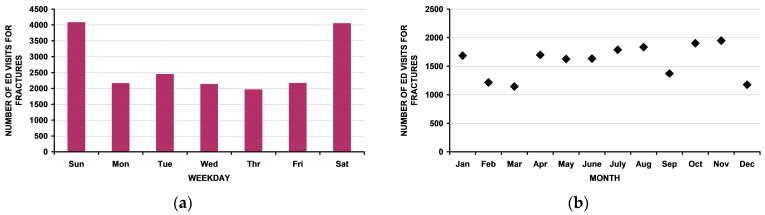
Temporal variability in the number of ED visits for fractures in children < 16 years old: (**a**) by day of the week. Note the increased number of ED visits on the weekend); (**b**) by month. Note that there is no apparent pattern. There was no change by month upon linear regression analysis: r^2^ = 0.057, *p* = 0.45. Additionally, cosinor analysis [37,38] demonstrated no rhythmic pattern as seen in many other pediatric non-firearm injuries [39,40,41]; (**c**) by year from 1993 through 2019. There was no change over time upon linear regression analysis: r^2^ = 0.05, *p* = 0.26.

**Table 1 children-10-00651-t001:** Demographics of the 19,033 patients.

	n	N	L95%CI	U95%CI	%
**All**	711	19,033	14,321	25,200	-
**Age (average in years)**	12.2 [11.7, 12.6]				
**Fracture location**					
**Spine**	39	717	416	1221	3.8
**Upper extremity**	292	9209	7612	10,823	48.4
**Lower extremity**	251	5982	4952	7118	31.5
**Skull/face**	114	2894	2137	3857	15.2
**Rib**	13	209	110	397	1.1
**Number of fractures per patient**					
**1**	695	18,640	18,245	18,834	98.0
**2**	13	365	173	760	1.9
**3**	1	6	0	42	0.0
**Sex**					
**Male**	586	16,203	15,578	16,731	85.2
**Female**	124	2825	2297	3450	14.8
**Race**					
**White**	214	8215	7032	9364	54.9
**Black**	270	4883	3584	6389	32.6
**Amerindian**	60	1809	933	3314	12.1
**Asian**	2	70	9	502	0.5
**Firearm type**					
**Powder**	536	12,314	10,662	13,801	64.7
**Non-powder**	175	6719	5232	8371	35.3
**Shot**					
**Yes**	617	15,376	14,378	16,203	80.8
**No**	94	3657	2830	4655	19.2
**Drive by shooting**			0	0	0.0
**Yes**	43	705	447	1100	3.7
**No**	668	18,328	17,933	18,586	96.3
**Disposition from ED**					
**Release**	385	13,272	11,267	15,334	68.7
**Admit**	321	5631	3889	7987	29.1
**Death**	3	100	21	485	0.5
**Who caused the injury**					
**Unknown**	207	4533	3624	5586	23.8
**Stranger**	91	1494	967	2273	7.8
**Self**	214	8148	6926	9419	42.8
**Friend/acquaintance**	58	1577	1146	2149	8.3
**Other relative**	54	1299	876	1907	6.8
**Other/not seen**	86	1906	1338	2678	10.0
**Injury intent**					
**Unknown**	37	985	560	1707	5.2
**Unintentional**	328	11,417	9977	12,771	60.0
**Assault**	336	6401	5097	7851	33.6
**Suicide**	4	157	34	689	0.8
**Law enforcement**	6	73	27	194	0.4
**Incident locale**					
**Unknown**	242	7190	5959	8502	37.8
**Home/apt**	272	7256	6001	8599	38.1
**School/recreation**	43	1298	862	1930	6.8
**Street/highway**	96	1867	1256	2727	9.8
**Other property**	57	1310	847	1997	6.9
**Farm**	1	112	15	797	0.6
**Year span**					
**1993–2001**	180	7027	5383	8848	36.9
**2002–2010**	252	6108	4762	7630	32.1
**2011–2019**	279	5898	4604	7370	31.0
**Hospital size**					
**Small**	64	5154	3493	7238	27.1
**Medium**	56	3485	2033	5632	18.3
**Large**	98	5226	2630	8982	27.5
**Very large**	227	3671	2282	5620	19.3
**Children**	266	1497	744	2893	7.9
**Argument**					
**Unknown**	306	7408	204,300	7321	69.5
**Yes**	29	691	405	1165	3.6
**No**	376	10,934	9606	12,208	57.4
**Crime**					
**Unknown**	290	6993	5836	8237	36.7
**Yes**	83	1208	700	2044	6.3
**No**	338	10,832	9505	12,109	56.9
**Drugs**					
**Unknown**	319	7665	6344	9064	40.3
**Yes**	17	398	188	830	2.1
**No**	375	10,970	9610	12,272	57.6
**Fight**					
**Unknown**	283	6606	5055	8350	34.7
**Yes**	34	705	390	1260	3.7
**No**	394	11,722	10,217	13,118	61.6

n = actual number, N = estimated number, L95%CI is the lower 95% confidence limit for N, U95%CI is the upper confidence limit for N.

**Table 2 children-10-00651-t002:** Anatomic distribution of 19,370 fractures in 19,033 patients.

	All Patients			Released from ED		Admitted to Hospital	
Bone	n	N	%N	N	N%	N	N%
**Finger**	157	6043	31.4%	5875	43.6%	168	2.9%
**Face**	70	2127	11.1%	1291	9.6%	836	14.5%
**Foot**	51	1394	7.2%	1186	8.8%	208	3.6%
**Toe**	33	1348	7.0%	1253	9.3%	95	1.6%
**Tibia/fibula**	75	1301	6.8%	316	2.3%	985	17.1%
**Hand**	46	1208	6.3%	1013	7.5%	195	3.4%
**Femur**	57	1036	5.4%	215	1.6%	821	14.2%
**Forearm**	37	844	4.4%	432	3.2%	412	7.1%
**Skull**	45	741	3.9%	217	1.6%	524	9.1%
**Humerus**	30	658	3.4%	476	3.5%	182	3.2%
**Ankle**	15	479	2.5%	374	2.8%	105	1.8%
**Knee**	12	399	2.1%	230	1.7%	169	2.9%
**Scapula/shoulder**	15	386	2.0%	168	1.2%	218	3.8%
**Cervical spine**	7	246	1.3%	120	0.9%	126	2.2%
**Thoracic spine**	15	219	1.1%	6	0.0%	213	3.7%
**Rib**	14	215	1.1%	65	0.5%	150	2.6%
**Lumbar spine**	12	168	0.9%	18	0.1%	150	2.6%
**Clavicle**	5	159	0.8%	143	1.1%	16	0.3%
**Sacrococcygeal spine**	5	84	0.4%	18	0.1%	66	1.1%
**Wrist**	7	81	0.4%	37	0.3%	44	0.8%
**Pelvis**	7	59	0.3%	6	0.0%	53	0.9%
**Elbow**	4	23	0.1%	11	0.1%	12	0.2%
**Hip**	2	22	0.1%	0	0.0%	22	0.4%

n = actual number, N = estimated number of ED visits.

**Table 3 children-10-00651-t003:** Analyses by fracture location.

	Spine					Upper Extremity					Lower Extremity					Skull/Face					Rib					*p* Value	*p* Value ^
Variable	n	N	L95%CI	U95%CI	%	n	N	L95%CI	U95%CI	%	n	N	L95%CI	U95%CI	%	n	N	L95%CI	U95%CI	%	n	N	L95%CI	U95%CI	%		
**All**	39	717	416	1221	4.0	292	9209	7612	10,823	12.8	251	5982	4952	7118	31.5	114	2894	2137	3857	15.2	13	209	110	397	1.1	-	-
**Age (average in years)**	11.6 [10.7, 12.5]					13.8 [12.2, 15.4]					12.7 [11.8, 13.6]					11.6 [10.7, 12.5]					13.8 [12.2, 15.4]					0.005	
**Age group (years)**																											
**≤5**	2	21	4	97	3	26	743	445	1212	8.1	16	429	141	1187	7.2	19	438	211	834	15.1	1	15	2	86	7	0.041	0.12
**6 to 10**	3	46	10	182	6	50	1802	1260	2504	19.6	27	633	398	982	10.6	17	336	178	603	11.6	0	0	0	0	0		
**11 to 15**	34	650	518	698	91	216	6664	5980	7250	72.4	208	4920	4318	5337	82.2	78	2120	1827	2357	73.3	12	194	123	207	93		
**Sex**																											
**Male**	33	596	481	660	84	250	8149	7609	8523	88.5	203	5142	4762	5418	86.0	87	2118	1714	2422	73.2	12	193	120	207	92	0.51	0.39
**Female**	5	116	52	231	16	42	1060	686	1600	11.5	48	840	564	1220	14.0	27	776	472	1180	26.8	1	16	2	89	8		
**Race**																											
**White**	5	133	37	330	23	114	4896	4134	5551	67.3	51	1808	1414	2236	39.1	40	1283	958	1584	56.9	4	95	38	131	66	0.003	0.0007
**Black**	19	277	90	479	47	97	1901	1335	2602	26.1	116	2131	1460	2834	46.1	31	504	291	810	22.3	5	48	12	105	34		
**Amerindian**	5	179	79	325	30	22	474	240	908	6.5	19	687	294	1432	14.9	14	469	190	968	20.8	0	0	0	0	0		
**Firearm Type**																											
**Powder**	37	705	666	714	98	168	4275	3460	5112	46.4	225	4841	4058	5354	80.9	92	2267	1841	2553	78.3	12	204	174	208	98	<10^−4^	0.0004
**Non-powder**	2	12	3	51	2	124	4934	4097	5749	53.6	26	1141	628	1924	19.1	22	627	341	1053	21.7	1	5	1	35	2		
**Shot**																											
**Yes**	38	701	602	715	98	238	6954	6238	7543	75.5	243	5720	5316	5883	95.6	83	1770	1376	2119	61.2	13	209	2	5	100	0.0001	<10^−4^
**No**	1	16	2	115	2	54	2255	1666	2971	24.5	8	262	99	666	4.4	31	1124	775	1518	38.8	0	0	0	0	0		
**Drive by shooting**																											
**Yes**	7	63	22	162	9	14	272	110	659	3.0	20	295	137	617	4.9	1	60	12	281	2.1	1	15	2	86	7	0.67	0.53
**No**	32	654	555	695	91	278	8937	8551	9099	97.0	231	5687	5365	5845	95.1	113	2834	2613	2882	97.9	12	194	123	207	93		
**Disposition from ED**																											
**Release**	6	162	44	405	23	220	8007	7347	8457	86.9	112	3525	2887	4116	59.1	42	1508	995	2004	52.4	4	65	23	130	31	0.0001	0.0001
**Admit**	33	555	312	673	77	72	1202	752	1862	13.1	138	2442	1851	3080	40.9	68	1271	755	1834	44.1	9	144	79	186	69		
**Death**	0	0	0	0	0	0	0	0	0	0.0			0	0	0.0	3	100	20	444	3.5			0	0	0		
**Who caused the injury**																											
**Unknown**	16	361	207	514	50	62	1658	1213	2221	18.0	79	1452	1004	2018	24.3	40	877	584	1238	30.3	9	168	110	196	80	<10^−4^	<10^−4^
**Stranger**	7	50	14	155	7	30	392	227	666	4.3	44	910	505	1548	15.2	8	132	41	402	4.6	1	5	1	35	2		
**Self**	2	21	4	98	3	128	5184	4338	5991	56.3	63	2259	1667	2919	37.8	21	684	479	943	23.6	0	0	0	0	0		
**Friend/acquaintance**	4	90	33	214	13	16	525	280	961	5.7	20	634	363	1069	10.6	17	323	127	739	11.2	1	5	1	35	2		
**Other relative**	3	58	14	202	8	19	475	222	984	5.2	11	244	100	574	4.1	20	506	308	792	17.5	1	16	3	72	8		
**Other/not seen**	7	137	72	240	19	36	899	541	1456	9.8	34	483	246	912	8.1	8	372	133	902	12.9	1	15	2	86	7		
**Injury intent**																											
**Unknown**	2	12	3	48	2	19	588	314	1072	6.4	13	264	112	599	4.4	3	121	31	432	4.2	0	0	0	0	0	<10^−^	<10^−4^
**Unintentional**	6	111	48	228	15	177	6747	5994	7377	73.3	93	3168	2408	3905	53.0	50	1370	899	1858	47.3	2	21	5	69	10		
**Assault**	30	578	474	645	81	93	1839	1244	2625	20.0	143	2528	1845	3264	42.3	57	1246	901	1615	43.1	11	188	140	204	90		
**Suicide**	0	0	0	0	0	0	0	0	0	0.0	0	0	0	0	0.0	4	157	41	542	5.4	0	0	0	0	0		
**Law enforcement**	1	16	2	105	2	3	35	10	116	0.4	2	22	5	102	0.4	0	0	0	0	0.0	0	0	0	0	0		
**Incident locale**																											
**Unknown**	15	254	136	404	35	99	3603	2789	4488	39.1	78	1879	1345	2511	31.4	44	1324	945	1720	45.7	5	113	53	168	54	0.003	0.002
**Home/apt**	11	218	153	297	30	127	3802	3020	4636	41.3	83	2162	1415	3040	36.1	46	1014	764	1296	35.0	4	55	22	109	26		
**School/recreation**	1	6	1	46	1	18	712	387	1270	7.7	14	302	166	540	5.0	8	258	88	677	8.9	2	20	4	83	10		
**Street/highway**	5	140	51	312	20	32	468	271	797	5.1	49	1080	720	1567	18.1	10	179	69	438	6.2	0	0	0	0	0		
**Other property**	7	99	44	203	14	16	624	288	1294	6.8	27	559	328	925	9.3	6	119	49	276	4.1	2	21	5	69	10		

n = actual number, N = estimated number, L95%CI is the lower 95% confidence limit for N, U95%CI is the upper confidence limit for N. ^ *p* value excluding the rib group.

**Table 4 children-10-00651-t004:** Analyses by firearm type.

	Powder					Non-Powder					
Variable	n	N	L95%CI	U95%CI	%	n	N	L95%CI	U95%CI	%	*p* Value
**All**	536	12,314	10,662	13,801	64.7	175	6719	5242	8382	35.3	-
**Age (average in years)**	12.5 [11.9, 13.1]					11.7 [11.1, 12.2]					0.58
**Age group (years)**											
**≤5 years**	53	1299	803	2047	10.5	12	352	76	335	5.2	0.012
**6 to 10**	56	1241	896	1698	10.1	41	1576	408	815	23.5	
**11 to 15**	427	9774	9006	10,402	79.4	122	4791	703	892	71.3	
**Sex**											
**Male**	425	10,012	9410	10,512	81.3	161	6191	5702	6456	92.1	0.0024
**Female**	110	2297	1797	2899	67.0	14	528	263	1017	7.9	
**Race**											
**White**	124	4336	3510	5195	45.7	90	3879	3166	4436	71.6	0.0005
**Black**	288	3740	2593	5032	39.4	32	1143	665	1832	21.1	
**Amerindian**	44	1413	803	2366	14.9	16	396	145	998	7.3	
**Fracture location**											
**Spine**	37	705	429	1140	5.8	2	12	3	45	0.2	<10^−4^
**Upper extremity**	168	4275	3327	5330	35.4	124	4934	4179	5528	73.5	
**Lower extremity**	225	4841	4070	5655	40.0	26	1141	648	1891	17.0	
**Head/face**	92	2267	1677	3005	18.8	22	627	330	1142	9.3	
**Shot**											
**Yes**	476	10,070	9244	10,712	81.8	141	5306	2830	4655	79.0	0.59
**No**	60	2244	1603	3070	18.2	34	1413	2830	4655	21.0	
**Drive by shooting**			0	0	0.0			0	0	0.0	
**Yes**	42	627	399	975	5.1	1	78	11	538	1.2	0.022
**No**	494	11,687	11,339	11,915	94.9	174	6641	6181	6708	98.8	
**Disposition from the ED**											
**Released**	233	6953	5669	8200	56.5	152	6319	2830	4655	94.0	<10^−4^
**Admitted**	298	5231	3987	6583	42.5	23	400	2830	4655	6.0	
**Died**	3	100	14	309	0.8	0	0	2830	4655	0.0	
**Who caused the injury**											
**Unknown**	181	3602	2890	4408	29.3	26	931	556	1498	13.9	<10^−4^
**Stranger**	90	1416	909	2154	11.5	1	78	11	538	1.2	
**Self**	106	3765	3045	4571	30.6	108	4383	3758	4938	65.2	
**Friend/acquaintance**	42	1176	815	1672	5.0	16	401	186	834	6.0	
**Other relative**	38	857	549	1319	7.0	16	442	237	802	6.6	
**Other/not seen**	79	1498	1058	2088	12.2	7	408	183	873	6.1	
**Injury intent**											
**Unknown**	26	504	249	1001	4.1	11	481	225	983	7.2	<10^−4^
**Unintentional**	173	5436	4536	6368	44.1	155	5981	5476	6297	89.0	
**Assault**	328	6151	5269	7034	50.0	8	250	101	600	3.7	
**Suicide**	3	150	33	659	1.2	1	7	1	51	0.1	
**Law enforcement**	6	73	28	191	0.6	0	0	0	0	0.0	
**Incident locale**											
**Unknown**	173	4422	5371	3553	36.2	69	2768	2057	3538	41.2	<10^−4^
**Home/apt**	177	3788	3010	4665	31.0	95	3468	2755	4171	51.6	
**School/recreation**	40	1206	772	1844	9.9	3	92	19	421	1.4	
**Street/highway**	93	1812	1300	2481	14.9	3	55	14	216	0.8	
**Other property**	52	974	638	1464	8.0	5	3336	123	868	49.7	
**Year group**											
**1993 to 2001**	109	3942	2756	5352	32.0	71	3085	2377	3819	45.9	0.006
**2002 to 2010**	187	3604	2601	4804	29.3	65	2504	1835	3255	37.3	
**2011 to 2019**	240	4768	3670	5969	38.7	39	1130	707	1734	16.8	

n = actual number, N = estimated number, L95%CI is the lower 95% confidence limit for N, U95%CI is the upper confidence limit for N.

**Table 5 children-10-00651-t005:** Analyses by being shot or not.

	Shot					Not Shot					*p* Value
Variable	n	N	L95%CI	U95%CI	%	n	N	L95%CI	U95%CI	%	
**All**	617	15,376	14,378	16,203	80.8	94	3657	2830	4655	19.2	-
**Age (average in years)**	11.5 [11.3, 11.6]					10.5 [10.1, 10.8]					<10^−4^
**Sex**											
**Male**	516	13,278	12,676	13,762	86.4	70	2925	2480	3231	80.0	0.27
**Female**	100	2093	1609	2695	13.6	24	732	426	1177	20.0	
**Race**											
**White**	168	6164	5167	7154	51.2	46	2051	1587	2396	71.7	0.036
**Black**	255	4389	3108	5851	36.4	15	494	266	853	17.3	
**Amerindian**	48	1494	772	2726	12.4	12	315	132	687	11.0	
**Firearm type**											
**Powder**	476	10,070	8534	11,421	65.5	60	2244	1769	2667	61.4	0.59
**Non-powder**	141	5306	3955	6842	34.5	34	1413	990	1888	38.6	
**Drive by shooting**											
**Yes**	43	705	452	1090	4.6	0	0	0	0	0.0	<10^−4^
**No**	574	14,671	14,286	14,924	95.4	94	3657	2830	4655	100.0	
**Disposition from ED**											
**Release**	298	9830	8049	11,390	64.1	87	3442	2988	3594	94.1	<10^−4^
**Admit**	314	5416	3846	7239	35.3	7	215	63	669	5.9	
**Death**	3	100	20	477	0.7	0	0	0	0	0.0	
**Fracture location**											
**Spine**	38	701	406	1195	4.6	1	16	2	116	0.4	<10^−4^
**Upper extremity**	238	6954	5658	8287	45.9	54	2255	1721	2722	61.7	
**Lower extremity**	243	5720	4798	6705	37.8	8	262	102	628	7.2	
**Skull/face**	83	1770	1213	2537	11.7	31	1124	721	1627	30.7	
**Who caused**	617	15,376	0	0	100.0	94	3657	0	0	100.0	
**Unknown**	185	3866	3089	4763	25.1	22	667	397	1062	18.2	0.18
**Stranger**	84	1366	864	2117	8.9	7	128	41	381	3.5	
**Self**	170	6151	5139	7222	40.0	44	1997	1526	2446	54.6	
**Friend/acquaintance**	53	1292	906	1822	5.0	5	285	106	707	7.8	
**Other relative**	44	1076	689	1656	7.0	10	223	88	535	6.1	
**Other/not seen**	81	1625	1147	2271	10.6	5	281	112	658	7.7	
**Injury intent**											
**Unknown**	36	951	533	1655	6.2	1	34	5	225	0.9	0.0005
**Unintentional**	268	8635	7417	9802	56.2	60	2782	2320	3121	76.1	
**Assault**	304	5565	4427	6809	36.2	32	836	507	1290	22.9	
**Suicide**	4	157	35	675	1.0	0	0	0	0	0.0	
**Law enforcement**	5	68	0	191	0.4	1	5	1	37	0.1	
**Incident locale**											
**Unknown**	203	5572	4573	6741	36.2	39	1618	1168	2095	44.2	<10^−4^
**Home/apt**	242	6200	5086	7477	40.3	30	1056	660	1566	28.9	
**School/recreation**	24	476	277	824	3.1	19	822	472	1323	22.5	
**Street/highway**	91	1775	1244	2526	11.5	5	92	26	312	2.5	
**Other property**	56	1241	823	1871	8.1	1	69	11	418	1.9	
**Age group (years)**											
**≤5**	56	1337	836	2093	8.7	9	314	138	674	8.6	0.95
**6 to 10**	87	2215	1691	2866	14.4	10	602	277	1175	16.5	
**11 to 15**	474	11,824	10,969	12,556	76.9	75	2741	2191	3134	75.0	

n = actual number, N = estimated number, L95%CI is the lower 95% confidence limit for N, U95%CI is the upper confidence limit for N.

**Table 6 children-10-00651-t006:** Analyses by disposition from the ED.

Variable	Released					Admitted					*p* Value
	n	N	L95%CI	U95%CI	%	n	N	L95%CI	U95%CI	%	
**All**	385	13,272	11,060	15,077	70.2	321	5631	3826	7843	29.8	-
**Age (average in years)**	12.1 [11.8, 12.7]					12.6 [11.9, 13.2]					0.29
**Age group (years)**											
**≤5**	33	1145	691	1851	8.6	29	406	53	170	7.2	0.74
**6 to 10**	51	2067	1449	2884	15.6	46	750	171	340	13.3	
**11 to 15**	301	10,060	9131	10,837	75.8	246	4475	1791	2100	79.5	
**Sex**											
**Male**	329	11,490	10,874	11,967	86.6	253	4598	4346	4810	81.7	0.12
**Female**	56	1782	1305	2398	67.0	1028	23	655	816	0.4	
**Race**											
**White**	141	6254	5183	7239	60.2	72	1945	1274	2670	43.9	0.011
**Black**	127	3091	2358	3941	29.8	141	1708	766	2893	38.6	
**Amerindian**	30	1036	530	1930	10.0	30	773	398	1380	17.5	
**Firearm Type**											
**Powder**	233	6953	5853	8035	52.4	298	5231	4936	5407	92.9	<10^−4^
**Non-powder**	152	6319	5237	7419	47.6	23	400	224	695	7.1	
**Shot**											
**Yes**	298	9830	8992	10,554	74.1	314	5416	2830	4655	96.3	<10^−4^
**No**	87	3442	2718	4280	25.9	7	215	2830	4655	3.8	
**Drive by shooting**											
**Yes**	17	399	186	841	3.0	26	396	199	465	7.0	0.15
**No**	368	12,873	12,431	13,086	97.0	295	5325	5166	5432	94.6	
**Fracture location**											
**Spine**	6	162	44	591	1.2	33	555	391	777	10.1	0.0001
**Upper extremity**	220	8007	7017	8935	60.6	72	1202	906	1562	22.0	
**Lower extremity**	112	3525	2660	4549	26.7	138	2442	2029	2869	44.6	
**Skull/face**	42	1508	1051	2128	11.4	68	1271	889	1754	23.2	
**Who caused the injury**											
**Unknown**	97	2723	2037	3568	20.5	108	1780	1391	2220	31.6	0.0006
**Stranger**	40	934	518	1644	7.0	51	560	264	1119	9.9	
**Self**	161	6922	5903	7927	52.2	50	1126	746	1636	20.0	
**Friend/acquaintance**	26	901	549	1453	5.0	32	676	427	1041	12.0	
**Other relative**	25	758	435	1297	5.7	29	541	361	798	9.6	
**Other/not seen**	35	958	617	1465	7.2	51	948	677	1299	16.8	
**Injury intent**											
**Unknown**	25	844	466	1492	6.4	12	141	52	368	2.5	<10^−4^
**Unintentional**	222	9195	8148	10,111	69.3	103	2122	1636	2657	37.7	
**Assault**	137	3228	2357	4293	24.3	197	3143	2593	3669	55.8	
**Suicide**	0	0	0	0	0.0	4	157	47	499	2.8	
**Law enforcement**	1	5	1	37	0.0	5	68	23	199	1.2	
**Incident locale**											
**Unknown**	142	5256	4210	6376	39.9	98	1904	1473	2389	33.8	0.02
**Home/apt**	148	5189	4177	6275	39.4	121	1967	1519	2468	34.9	
**School/recreation**	31	1104	697	1715	8.4	12	194	111	334	3.4	
**Street/highway**	40	762	455	1255	5.8	56	1105	849	1415	19.6	
**Other property**	23	849	458	1532	6.5	34	461	338	622	8.2	
**Year group**											
**1993 to 2001**	132	5692	1597	6891	42.9	48	1335	856	1971	23.7	0.004
**2002 to 2010**	132	4251	3331	5290	32.0	118	1826	1265	2493	32.4	
**2011 to 2019**	121	3329	2483	4348	25.1	155	2470	1879	3094	43.9	

n = actual number, N = estimated number, L95%CI is the lower 95% confidence limit for N, U95%CI is the upper confidence limit for N.

**Table 7 children-10-00651-t007:** Compilation of the literature regarding firearm-associated fractures in children.

Study	Year	Location	Number of Patients with Fractures	Inpatient or Outpatient	% Male	Age Limit (years)	% UE	% LE	% Spine
**Present**	2023	Entire USA	12,314	Both	81	<16	42.5	52.0	5.5
**Blumberg et al.** [14]	2018	Entire USA	2814	IP	91	<21	30.3	59.1	18.9
**Naranje et al.** [19]	2016	Birmingham, AL and Memphis, TN	49	IP	84	<19	22	76	2
**Perkins** [20]	2016	Charlotte, NC	44	IP	78	<18	41	59	-
**Washington et al.** [25]	1995	Los Angeles, CA	29	IP	81	<18	55	44	-
**Victoroff et al.** [23]	1994	Washington, DC	23	IP	83	<19	48	52	-
**Stucky et al.** [24]	1991	Detroit, MI	44	IP	83	<18	50	50	-

UE = upper extremity, LE = lower extremity.

## Data Availability

This data is freely to anyone online at the Inter-University Consortium for Political and Social Research Firearm Injury Surveillance Study 1993–2020 (ICPSR 38574) (https://www.icpsr.umich.edu/web/NACJD/studies/38574, accessed on 18 December 2022).

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
