# Peer review of "Fractures in Children Due to Firearm Activity"

_children, 2023, doi:10.3390/children10040651_

Round 1
Reviewer 1 Report
Dear Authors,
Thank you for the opportunity to review this article.
There are a lot of English errors, such as Children’s instead of Childrens. English revision must be conducted.
In the introduction, you should explain more about the two types of firearms (powder vs. non-powder), because not every physician who reads the paper could know what that means. Also, if possible, you should differentiate between recreational firearms and self-defense apparel, even when used abusively (I.E. mass shootings). Also, the mechanism of fracture is important. Is it a direct gunshot fracture, or an indirect mechanism from falling or tripping after being shot? High-energy trauma brings totally different health costs compared to low-energy trauma.
In this phrase „Over the 27-year period 1993 through 2019, there were 111,796 actual ED visits for 117 injuries due to firearms, for an estimated 3,359,809” you mention a number of visits, and after that, you elaborate on the second number (3.36 million ED visits). It is a bit ambiguous. How many ED visits were in that interval? 111796 or 3.36 million?
In this phrase „Children who sustained injuries at schools and recreational facilities were less likely to be shot compared to other places” can you give examples of the different ways they can injure themselves with the firearm without being shot?
A time-dependent graph could be useful for the public health domain to see the evolution of firearm injuries regarding incidence and severity. A more precise estimation of costs and disease burden could raise the value of your study.
Reviewer 2 Report
The study addresses an interesting topic: firearm fractures in children have not only a significant impact on global healthcare spending, but they are also a trauma for families.
Abstract’s structure is not properly organized: authors should introduce better the topic of the study and summarize, in an organized way, the different parts of the article. The aim of the study is clear, but authors should add it in the final part of the abstract.
The introduction is acceptable. According to the authors, what is the strength of the article compared to others?
The methodological approach is correct. All data are correctly reported.
Results are well presented. Table and charts are clear and exhaustive. All data are detailed and result correct in all sections.
Discussion is overall well written: all references are correctly reported.
Conclusions are acceptable.
Round 2
Reviewer 1 Report
The authors have made the modifications requested.
It's ok now.
Thank you